# Detection of coinfection with *Primate Erythroparvovirus 1* and arboviruses (DENV, CHIKV and ZIKV) in individuals with acute febrile illness in the state of Rio Grande do Norte in 2016

**Vanessa dos Santos Morais**[1]☺*, **Lídia Maria Reis Santana**[2,3], **João Felipe Bezerra**[4], **Flavia Emmanuelle Cruz**[5], **Themis Rocha de Souza**[6], **Roozbeh Tahmasebi**[1], **Rafael Augusto Alves Raposo**[1], **Roberta Marcatti**[1], **Erick Matheus Garcia Barbosa**[1], **Philip Michael Hefford**[7], **Renata Buccheri**[8], **Ester Cerdeira Sabino**[1], **Antonio Charlys da Costa**[1]☺

1 University of Sao Paulo/Department of Infectious and Parasitic Diseases, Sao Paulo, Brazil, 2 Sao Paulo Health Department/Epidemiological Surveillance Center "Prof. Alexandre Vranjac", Sao Paulo, Brazil, 3 Federal University of São Paulo, Sao Paulo, Brazil, 4 Federal University of Paraiba, Paraiba, Brazil, 5 Guamare Health Department, Guamare, Brazil, 6 Central Public Health Laboratory of the State of Rio Grande do Norte, Natal, Brazil, 7 University Hospitals Sussex NHS Foundation Trust, Chichester, England, 8 Vitalant Research Institute, Department of Epidemiology, San Francisco, California, United States of America

☺ These authors contributed equally to this work.
* va.morais@usp.br

**Data Availability Statement:** All data are in the manuscript and/or supporting information files.

## Abstract

### Background

Arthropod-borne viruses, known as arboviruses, pose substantial risks to global public health. Dengue (DENV), Chikungunya (CHIKV) and Zika (ZIKV) viruses stand out as significant concerns in Brazil and worldwide. Their overlapping clinical manifestations make accurate diagnosis a challenge, underscoring the need for reliable laboratory support. This study employs a comprehensive molecular diagnostic approach to track viral infections in individuals with acute febrile illness, a period marked by widespread outbreaks of DENV, CHIKV and ZIKV.

### Methods

Between January and August 2016, we received a total of 713 serum samples obtained from individuals with acute febrile illness, previously tested for DENV, CHIKV or ZIKV, with initial negative results, from LACEN-NATAL. Of the total 713 samples, 667 were from females (354 of them pregnant) and 46 from males. Molecular diagnosis was conducted using the Multiplex RT-qPCR technique for simultaneous detection of DENV, CHIKV and ZIKV. Additionally, we performed differential diagnosis by RT-qPCR for other viruses of the Flavivirus, Alphavirus Enterovirus genera and qPCR for Primate Erythroparvovirus 1 (B19V) species, in accordance with Ministry of Health guidelines.

**Funding:** This work was developed with the support of the São Paulo Research Foundation - FAPESP, Centre for Arbovirus Discovery, Diagnostics, Genomics & Epidemiology (CADDE) #2018/14389-0. VdSM is supported by FAPESP #2019/21706-5. The funders had no role in the study design, data collection and analysis, decision to publish, or manuscript preparation.

**Competing interests:** The authors have declared that no competing interests exist.

## Results

Among the 713 cases, 78.2% tested positive for viral infections, including 48% with CHIKV viremia, 0.6% with DENV and 0.1% with ZIKV. Arboviral coinfections totaled 2.4%, including DENV-CHIKV (1.7%) and CHIKV-ZIKV (0.7%). Moreover, 8% exhibited B19V viremia. Simultaneous infections were identified in 17.5%, encompassing B19V-CHIKV (17.1%), B19V-DENV (0.1%), and B19V-ZIKV (0.3%) Triple infections were observed in 1.3% of cases with B19V-DENV-CHIKV (1%) and B19V-CHIKV-ZIKV (0.3%).

## Conclusion

Molecular testing demonstrated high efficacy in diagnosing prevalent arboviruses and detecting multiple coinfections. This approach helps to elucidate etiologies for symptomatic cases, especially during arbovirus outbreaks, and aids comprehensive surveillance. Our findings underscore the importance of monitoring co-circulating pathogens, such as B19V, with implications for clinical management, particularly in pregnant individuals. This study enhances our understanding of arbovirus epidemiology and reinforces the critical role of molecular diagnosis in disease surveillance and control.

## Author summary

Fever is one of the most common reasons for seeking health care in Latin America and is a substantial contributor to continental morbidity and mortality. Viruses transmitted by arthropods, known as arboviruses, represent one of the predominant etiologic agents responsible for causing febrile illness in Latin America. In the current epidemiological scenario in Brazil, due to shared characteristics of DENV, CHIKV and ZIKV, the Ministry of Health recommends performing a differential diagnosis for other clinical syndromes. In this study, we performed the differential molecular diagnosis in 713 samples of individuals with acute febrile illness suspected of arboviruses belonging to the state of Rio Grande do Norte in 2016. The samples were submitted to RT-qPCR and qPCR tests for DENV, CHIKV and ZIKV, together with other agents responsible for causing febrile syndrome and exanthematous fever syndrome. These include Enterovirus, other arboviruses of the Flavivirus and Alphavirus genera and the species Primate Erythroparvovirus 1 (B19V). We report the concomitant detection of CHIKV and B19V in 17.1% of cases, highlighting the importance of a broad diagnostic approach, since the detection of simultaneous infections may be occurring between arboviruses and/ or other viruses during epidemics in different regions of Brazil.

## Introduction

Brazil is known for the circulation of several arboviruses of medical importance, including Dengue and Zika (DENV and ZIKV, genus *Flavivirus*) and chikungunya (CHIKV, genus *Alphavirus*). These 3 viruses share common hosts and vectors and undergo periodic amplifications, during which they are detectable in the population causing an acute febrile illness, with clinically similar characteristics. Among the symptoms, the most common include fever, rash, muscle pain, arthralgia, and headaches [1].

Arboviruses are notifiable diseases at the national level in Brazil. However, in the public sector which serves more than 70% of the Brazilian population, only a small proportion of cases reported by health units are submitted for confirmation testing in public reference laboratories via specific molecular and/or serological assays, or virus culture. In 2020, among DENV cases reported up to April, only 23% were tested in reference laboratories [2]. Rapid tests, especially for DENV, are typically effective only up to the fifth day post-infection, as a high error rate can accrue beyond this period. There is therefore reliance upon the use of molecular and/or serological laboratory tests which requires technical equipment not widely available throughout Brazil. Furthermore, for DENV and ZIKV, cross-reactivity of serological assays represents a serious problem, leading to erroneous results. [3]. In Brazil, the Ministry of Health recommends that samples sent to perform the RT-qPCR test for DENV, CHIKV and ZIKV should be collected up until the fifth day of the onset of symptoms. The IgM/IgG method, in contrast, must be collected from the sixth day of symptom onset [4]. Due to the difficulty of differentiating arbovirus species based only on clinical and epidemiological symptoms; laboratory tests are necessary for an accurate diagnosis. In addition, a broad diagnostic approach helps in the detection of concomitant arboviral infections or other viruses with similar symptoms, since we have coexisting circulation of these agents, which may have relevant clinical implications [5].

Rio Grande do Norte is a Brazilian state located in the Northeast region of the country, bordering to the North and East with the Atlantic Ocean, to the South with the state of Paraiba and to the West with Ceara. According to the Brazilian Institute of Geography and Statistics (IBGE), in 2020, the state had a population estimate of 3,534,165 million inhabitants, composed of 77.81% urban residents and 22.19% rural residents. Its human development index is 0.684 [6]. The State Department of Public Health (SESAP) reported an increase in the number of cases of DENV, CHIKV and ZIKV infection in 2016 compared to 2015 [4], demonstrating that these viruses circulated simultaneously throughout the state.

Our study describes the laboratory results of individuals with acute febrile syndromes reported with suspected arboviral diseases to the Notifiable Diseases Information System (SINAN), during the period from January to August 2016. We emphasize the emergence and reemergence of arboviral diseases, particularly in regions with concurrent circulation. Additionally, we demonstrate molecular approaches that contribute to a rapid and accurate differential diagnosis of DENV, CHIKV and ZIKV, alongside their implications for other viruses of the Flavivirus, Alphavirus and Enterovirus genera and for the Primate Erythroparvovirus 1 (B19V) species of the Erythroparvovirus genus.

## Methodology

**Ethics statement.**   The study was approved by the Committee for Ethics in Research of the School of Medicine of the University of São Paulo, CAAE: 53153916.7.0000.0065. All participants enrolled signed an informed written consent form to participate in the study.

This cross-sectional observational study utilized serum samples obtained from patients with acute febrile syndrome consulted at health facilities between January 1 and August 31, 2016, in the state of Rio Grande do Norte. The clinical and epidemiological data of the patients were obtained through consultations with the Notifiable Diseases Information System (SINAN), which is fed mainly by the notification and investigation of cases of diseases and conditions that appear in the national list of diseases of compulsory notification. It is an important tool to assist in health planning, define intervention priorities, and allow the impact of interventions to be evaluated [7].

Based on SINAN data, a total of 86,628 suspected cases of arboviruses were reported during 2016, of which 56,471 referred to as DENV, 26,560 as CHIKV and 3,597 as ZIKV [4].

Among the 12,163 samples of suspected arboviruses that were sent to the Central Public Health Laboratory (LACEN-NATAL) during the study period, 126 were requested for molecular diagnosis of DENV, 77 samples were requested for molecular diagnosis for CHIKV, and 510 samples were requested for molecular diagnosis for ZIKV; totalling 713 samples with a previous negative molecular diagnosis. The 713 samples were sent to the Instituto de Medicina Tropical da Faculdade de Medicina (IMT) for molecular analysis through RT-qPCR and qPCR methodology and subsequent viral metagenomics in samples with negative results for the tested viruses. All samples tested were within 5 days of symptom onset, following criteria established by the World Health Organization (WHO) [8] and Centre for Disease Control and Prevention (CDC) [9] and were collected for the purpose of laboratory investigation of DENV, CHIKV and/or ZIKV as established by the epidemiological surveillance of these diseases.

## Molecular analysis

The total amounts of nucleic acids were extracted from 200ul aliquots of human serum samples using the QIAamp MinElute Virus Spin Kit (Qiagen), following the manufacturer's instructions. The eluate containing 60ul of total RNA and DNA was stored in a -80˚C freezer until ready for use. For the RT-qPCR assay, we used the Kit Molecular ZDC–BioManguinhos (Instituto de Tecnologia em Imunobiológicos, Fundação Oswaldo Cruz–RJ–Brasil). Reverse transcription of RNA into cDNA preceded the amplification. The target-specific amplification methodology using fluorescence-labeled probes was employed to determine the arboviral presence; discriminative for DENV, CHIKV and ZIKV. The Molecular Kit was standardized to detect truly positive DENV samples with Ct values less than or equal to 33, and for truly positive CHIKV and ZIKV samples with Ct values less than or equal to 38. In all reactions, the negative and positive controls provided in the Kit were used. All samples were initially tested simultaneously for DENV, CHIKV and ZIKV, followed by differential diagnosis.

For Flavivirus genus detection we followed the protocol established by Patel et al. [10]. Cell culture samples of the four DENV serotypes and Saint Louis Encephalitis virus (SLEV) were used as positive controls for this assay. For Alphavirus genus detection we followed the protocols established by Giry et al. [11]. Cell culture samples of Mayaro virus (MAYV) and CHIKV were used as positive controls for this assay. For Enterovirus genus detection we followed the protocol established by Hymas et al. [12]. Positive controls used were derived from cell culture samples of Echovirus 30. For B19V species detection, the protocol of Jia J et al. [13] was followed. Positive controls used in this assay were obtained from known positive samples stored in the laboratory. The processing of all samples included negative controls as well as internal controls to ensure the reliability of the reaction. The reactions were conducted using the ABI 7500 real-time PCR system.

## Database

Probable cases of DENV, CHIKV and ZIKV- defined as cases reported as suspected for each of these arboviruses minus those considered to be discarded, where symptoms started in 2016 and the municipality of residence was in the state of Rio Grande do Norte (RN)—come from the Notifiable Diseases Information System (SINAN). The population residing in RN by municipality comes from the 2010/IBGE Census.

## Used software

Construction and analysis of databases, elaboration of graphs and maps were conducted using the dplyr and ggplot2 packages in the R environment version 4.2.2 (2022-10-31) [14–16].

## Results

### Arboviruses

Between January and December 2016, 86,628 probable cases of arboviruses were reported, of which 65,2% were for DENV, 30,6% for CHIKV and 4.2% for ZIKV, in state residents of Rio Grande do Norte. The mean annual incidences of probable cases of DENV, CHIKV, and ZIKV during the study period were, respectively, 1782.5, 838.4, and 113.5 per 100,000 population, with the highest rates observed between February and May 2016 (**Fig 1**).

The spatial distribution of arboviruses was quite heterogeneous among the municipalities. The northeast, south and southwest regions of the state of RN, which represent most of the municipalities, had the highest incidences of probable DENV cases, while for CHIKV the most affected areas were southwest, south and northwest. ZIKV cases were more frequently identified in central regions of the state and most municipalities had a low incidence (**Fig 2**).

The mean age was higher among CHIKV suspects and lower among DENV and ZIKV suspects, with a higher percentage of cases in the age groups above 30 years for probable CHIKV (18.1%), while ZIKV and DENV suspects were more concentrated in the age groups up to 29 years (25.2%), (16.7%), respectively. There was a predominance of females in all groups, more prominent among ZIKV suspects (69.9% of women), which also had a higher proportion of pregnant women compared to the other groups (22.8% of women), and less for DENV (57.9% of women, 3.1% of whom were pregnant). The most informed race/color was brown, varying between 62.1% for DENV, 59.2% for CHIKV and 64.6% for ZIKV.

The laboratory confirmations are consistent with the clinic-epidemiological criteria, particularly among suspected cases of CHIKV (6.3%), followed by DENV (2.2%) and ZIKV (0.7%).

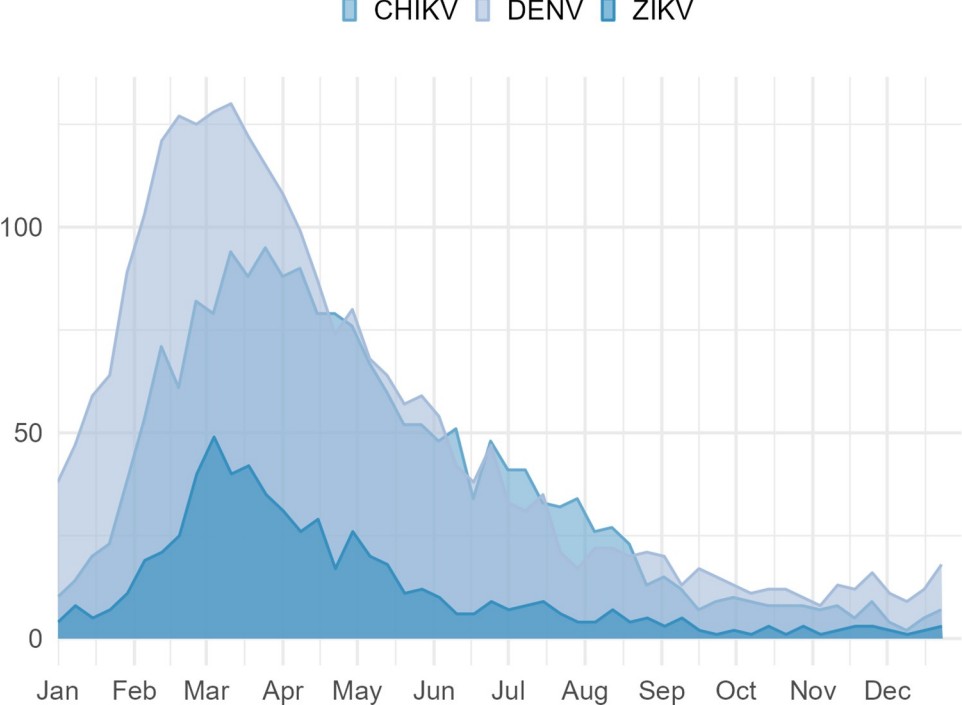

**Fig 1. Average annual incidence rates of probable cases of DENV, CHIKV and ZIKV, RN, 2016.** The figure was created using the ggplot2 package in R version 4.2.2. The data for dengue, chikungunya, and Zika numbers were retrieved from the SINAN Database, while population numbers, used for the incidence calculations, were obtained from the Brazilian Institute of Geography and Statistics (Instituto Brasileiro de Geografia e Estatística—IBGE).

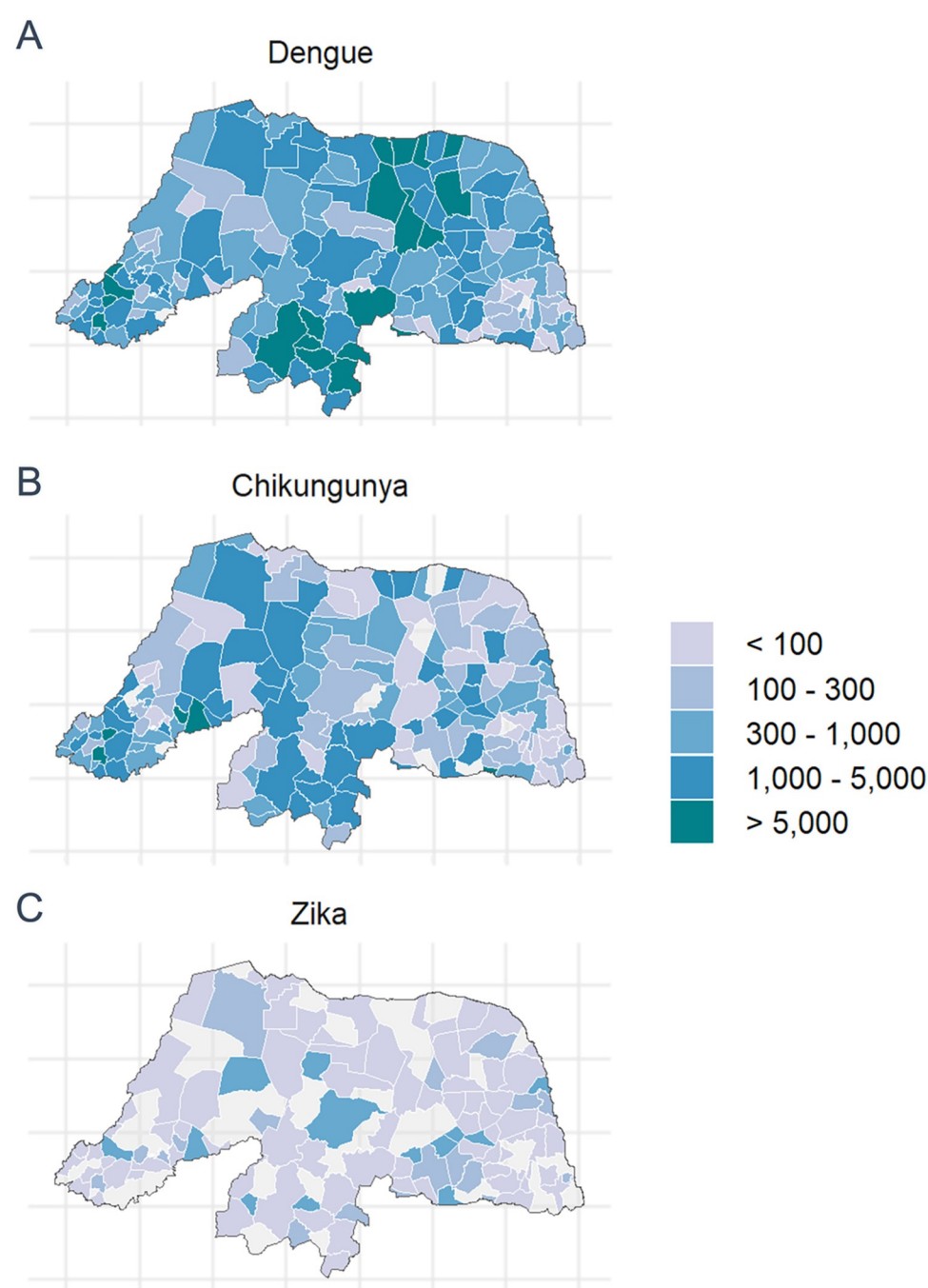

**Fig 2.** Incidence of probable cases of Dengue (Fig 2A), Chikungunya (Fig 2B), Zika (Fig 2C) per 100,000 inhabitants according to municipality of residence, RN, 2016. Maps were created using the ggplot2 package in R version 4.2.2. The shapefile for the RN map was downloaded from the Instituto Brasileiro de Geografia e Estatística–IBGE website (https://www.ibge.gov.br/geociencias/organizacao-do-territorio/malhas-territoriais/15774-malhas.html). The data for Dengue, Chikungunya, and Zika numbers were retrieved from the SINAN Database, while population numbers, used for the incidence calculations, were obtained from the Brazilian Institute of Geography and Statistics (Instituto Brasileiro de Geografia e Estatística—IBGE).

**Table 1. Description of probable cases of DENV, CHIKV and ZIKV notified in SINAN, RN, 2016.**

| | DENV | | CHIKV | | ZIKV | |
|---|---|---|---|---|---|---|
| | 56471 | | 26560 | | 3597 | |
| Total (N = 86628) | n | (%) | n | (%) | n | (%) |
| **Gender** | | | | | | |
| Male | 23785 | (42.0) | 9901 | (37.0) | 1083 | (30.0) |
| Female | 32686 | (58.0) | 16659 | (63.0) | 2514 | (70.0) |
| Pregnant | 998 | (1.8) | 1069 | (4.0) | 573 | (15.9) |
| **Age group** | | | | | | |
| 0–9 | 5731 | (10.3) | 1824 | (7.2) | 453 | (13.3) |
| 10–19 | 8411 | (15.2) | 2909 | (11.4) | 519 | (15.3) |
| 20–29 | 10420 | (18.8) | 4329 | (17.0) | 855 | (25.2) |
| 30–39 | 9278 | (16.7) | 4620 | (18.1) | 657 | (19.3) |
| 40–49 | 7304 | (13.2) | 3922 | (15.4) | 382 | (11.2) |
| 50–59 | 5997 | (10.8) | 3423 | (13.4) | 266 | (7.8) |
| 60–69 | 4063 | (7.3) | 2268 | (8.9) | 160 | (4.7) |
| 70–79 | 2796 | (5.0) | 1459 | (5.7) | 63 | (1.9) |
| 80–89 | 1216 | (2.2) | 587 | (2.3) | 35 | (1.0) |
| 90+ | 272 | (0.5) | 119 | (0.5) | 9 | (0.3) |
| **Self-reported race** | | | | | | |
| White | 7134 | (32.6) | 4731 | (35.2) | 402 | (28.3) |
| Black | 842 | (3.8) | 608 | (4.5) | 84 | (5.9) |
| Brown | 13588 | (62.1) | 7955 | (59.2) | 918 | (64.6) |
| Indigenous | 95 | (0.4) | 50 | (0.4) | 4 | (0.3) |
| Asian/Native/Other | 223 | (1.0) | 103 | (0.8) | 12 | (0.8) |
| **Clinical signs** | | | | | | |
| Fever | 5256 | (83.5) | 4012 | (87.7) | - | - |
| Myalgia | 3701 | (58.8) | 2948 | (64.5) | - | - |
| Headache | 4215 | (67.0) | 2987 | (65.3) | - | - |
| Rash | 1782 | (28.3) | 1466 | (32.1) | - | - |
| Vomit | 1438 | (22.8) | 1000 | (21.9) | - | - |
| Nausea | 1711 | (27.2) | 1383 | (30.2) | - | - |
| Back pain | 1392 | (22.1) | 1791 | (39.2) | - | - |
| Conjunctivitis | 253 | (4.0) | 442 | (9.7) | - | - |
| Arthritis | 845 | (13.4) | 1634 | (35.7) | - | - |
| Arthralgia | 2251 | (35.8) | 3333 | (72.9) | - | - |
| Petechiae | 476 | (7.6) | 350 | (7.7) | - | - |
| Leukopenia | 116 | (1.8) | 62 | (1.4) | - | - |
| Tourniquet test | 45 | (0.7) | 23 | (0.5) | - | - |
| Retro orbital pain | 914 | (14.5) | 496 | (10.8) | - | - |
| **Diagnostics criteria** | | | | | | |
| Clinico—epidemiological | 8555 | (15.1) | 8917 | (33.6) | 197 | (5.5) |
| Laboratory confirmation | 1258 | (2.2) | 1674 | (6.3) | 25 | (0.7) |
| Ignored | 46658 | (82.6) | 15969 | (60.1) | 3375 | (93.8) |

Abbreviations: DENV, dengue virus; CHIKV, chikungunya virus; ZIKV, zika virus

Cases classified as undetermined (without diagnosis) exhibited a higher percentage among suspected cases of ZIKV (93.8%), while suspected cases of DENV and CHIKV had 82.6% and 60.1%, respectively (**Table 1**).

**Table 2. Description of probable cases of DENV, CHIKV and ZIKV in pregnant women reported in SINAN, RN, 2016.**

| | DENV | | CHIKV | | ZIKV | |
|---|---|---|---|---|---|---|
| | 998 | | 1069 | | 573 | |
| Total (N = 2640) | n | (%) | n | (%) | n | (%) |
| **Age group** | | | | | | |
| 10–19 | 212 | (22.6) | 183 | (18.6) | 106 | (19.4) |
| 20–29 | 469 | (50.1) | 495 | (50.3) | 279 | (51.2) |
| 30–39 | 234 | (25.0) | 267 | (27.1) | 142 | (26.1) |
| 40–49 | 22 | (2.3) | 40 | (4.1) | 18 | (3.3) |
| **Self-reported race** | | | | | | |
| White | 237 | (30.8) | 302 | (33.4) | 142 | (29.8) |
| Black | 35 | (4.5) | 54 | (6.0) | 20 | (4.2) |
| Brown | 480 | (62.3) | 537 | (59.4) | 311 | (65.2) |
| Indigenous | 1 | (0.1) | 2 | (0.2) | 0 | (0.0) |
| Asian/Native/Other | 17 | (2.2) | 9 | (1.0) | 4 | (0.8) |
| **Clinical signs** | | | | | | |
| Fever | 167 | (93.8) | 110 | (77.5) | - | - |
| Myalgia | 143 | (80.3) | 92 | (64.8) | - | - |
| Headache | 142 | (79.8) | 91 | (64.1) | - | - |
| Rash | 89 | (50.0) | 61 | (43.0) | - | - |
| Vomit | 37 | (20.8) | 32 | (22.5) | - | - |
| Nausea | 51 | (28.7) | 39 | (27.5) | - | - |
| Back pain | 68 | (38.2) | 75 | (52.8) | - | - |
| Conjunctivitis | 12 | (6.7) | 11 | (7.7) | - | - |
| Arthritis | 59 | (33.1) | 53 | (37.3) | - | - |
| Arthralgia | 80 | (44.9) | 76 | (53.5) | - | - |
| Petechiae | 27 | (15.2) | 27 | (15.2) | - | - |
| Leukopenia | 3 | (1.7) | 2 | (1.4) | - | - |
| Tourniquet test | 1 | (0.6) | 1 | (0.7) | - | - |
| Retro orbital pain | 44 | (24.7) | 22 | (15.5) | - | - |
| **Diagnostics criteria** | | | | | | |
| Clinico—epidemiological | 151 | (15.1) | 158 | (14.8) | 16 | (2.8) |
| Laboratory confirmation | 16 | (1.6) | 118 | (11.0) | 3 | (0.5) |
| Ignored | 831 | (83.3) | 793 | (74.2) | 554 | (96.7) |

Abbreviations: DENV, dengue virus; CHIKV, chikungunya virus; ZIKV, zika virus

A total of 2,640 suspected cases of arboviral infections were recorded in pregnant women within the SINAN database, of which 38% were associated with DENV, 40.5% with CHIKV and 21.7% with ZIKV. Regarding suspected cases of DENV, CHIKV and ZIKV, the age group with the highest observed percentage was between 20 and 29 years; with 50.3% of suspected cases positive for CHIKV, 50.1% for DENV and 51.2% for ZIKV (**Table 2**).

## Molecular analyses

Out of the 713 cases analyzed, 78.2% showed positivity in the molecular diagnoses performed. Thus, 48% showed viremia for CHIKV, 0.6% tested positive for DENV, and 0.1% for ZIKV. A total of 2.4% (n = 17) arboviral coinfections were identified, with 1.7% (n = 12) of individuals coinfected with DENV and CHIKV, and 0.7% (n = 5) showing coinfection with CHIKV and ZIKV. The 713 individuals subjected to Flavivirus and Alphavirus genera investigation

**Table 3. Molecular analyses in samples from individuals with acute febrile illness reported as suspected arboviruses, RN, 2016.**

| Positive samples | | | | | n = 558 | % |
|---|---|---|---|---|---|---|
| DENV cases | | | | | 4 | 0.6 |
| CHIKV cases | | | | | 344 | 48.2 |
| ZIKV cases | | | | | 1 | 0.1 |
| B19V cases | | | | | 58 | 8.1 |
| DENV and CHIKV cases | | | | | 12 | 1.7 |
| CHIKV and ZIKV cases | | | | | 5 | 0.7 |
| B19V and CHIKV cases | | | | | 122 | 17.1 |
| B19V and ZIKV cases | | | | | 2 | 0.3 |
| B19V and DENV cases | | | | | 1 | 0.1 |
| B19V, DENV and CHIKV cases | | | | | 7 | 1 |
| B19V, CHIKV and ZIKV cases | | | | | 2 | 0.3 |

Abbreviations: DENV, dengue virus; CHIKV, chikungunya virus; ZIKV, zika virus; B19V, primate erythroparvovirus 1.

presented outcomes that align with those obtained in the Multiplex ZDC–BioManguinhos assay. In contrast, Enterovirus genus results were negative. We also uniformly assessed the presence of B19V among study participants, demonstrating a total of 8.0% (n = 58) positive samples for B19V. Furthermore, 17.5% (n = 125) concomitant infections with arboviruses were identified, with 17.1% (n = 122) of coinfections by B19V and CHIKV, 0.3% (n = 2) of coinfections by B19V and ZIKV, and 0.1% (n = 1) of coinfection by B19V and DENV. Triple infections were also identified in 1.3% (n = 9) of individuals, with 1.0% (n = 7) showing B19V, DENV and CHIKV coinfection and 0.3% (n = 2) displaying B19V, CHIKV and ZIKV coinfection (Table 3).

Out of the 713 individuals in the study, 50.0% (n = 354) were pregnant women, with 60.0% of them having arboviral infections, of which 57.0% (n = 203) were positive for CHIKV, 1.0% (n = 4) were positive for DENV and 1.7% (n = 6) had co-infection with DENV and CHIKV. Similarly, we identified 21.0% of B19V infections, with 4.5% (n = 16) having mono-infection, 16.0% (n = 57) having co-infections with B19V and CHIKV and 0.3% (n = 1) having triple infections with B19V, DENV and CHIKV. Of the total samples analyzed, 21.7% had negative results for all molecular tests.

The months with the highest number of collections for molecular analysis were February and March, coinciding with those with the highest incidence of probable cases of arboviruses (Fig 3).

## Discussion

Arboviruses are responsible for large epidemics of neglected tropical diseases annually in Brazil. In this study, we demonstrate the simultaneous detection of DENV, CHIKV and ZIKV arboviruses, and report infections caused by B19V during the spread of ZIKV and CHIKV in 2015 and 2016, respectively, in the state of Rio Grande do Norte.

The 713 samples analysed in our study showed a predominance of female infection (93.5%), 53% of which were pregnant women, with a higher proportion of suspected ZIKV patients explaining the clinical and laboratory hypothesis initially carried out in the study samples, which prioritised the diagnosis for ZIKV of this group. In our analyses, we obtained 48% positivity for CHIKV, 0.6% for DENV, 0.1% for ZIKV, as well as concomitant infections with each other; with 1.7% for CHIKV and DENV and 0.7% for CHIKV and ZIKV. Our findings

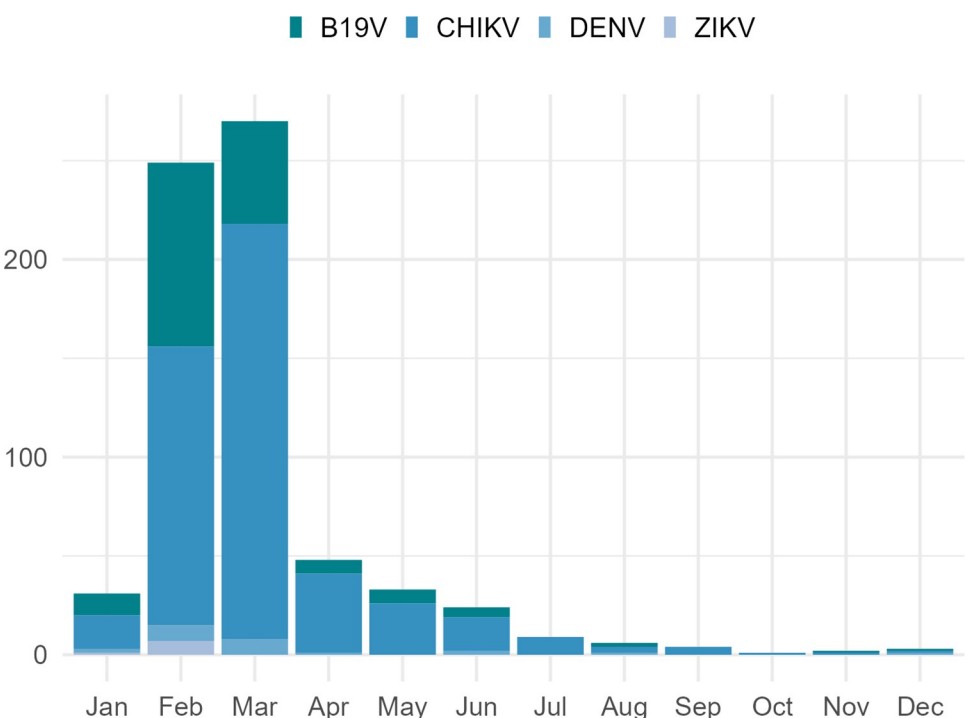

**Fig 3. Results of molecular analyses on samples from individuals with acute febrile illness reported as suspected arboviruses, according to month of collection, during 2016 in Rio Grande do Norte.** The figure was created using the ggplot2 package in R version 4.2.2.

corroborate the information presented in the 2016 Epidemiological Bulletin [4], for these arboviruses, highlighting the CHIKV infections that were predominant in the state of Rio Grande do Norte during the study period.

Despite detecting and reporting cases of arboviruses in approximately 50% of the individuals studied, we followed the Ministry of Health's recommendations and conducted a differential diagnosis on the 713 study samples. After identifying DENV, CHIKV and ZIKV, we proceeded to test for other arboviruses of the Flavivirus, Alphavirus and Enterovirus genera, as well as for B19V species, which also represent important pathogens causing febrile and exanthematic diseases. In our analysis, there were no additional identifications of viruses belonging to the Flavivirus and Alphavirus genera to the DENV, CHIKV and ZIKV already described. Similarly, we were unable to detect additional viruses of the Enterovirus genus.

Surprisingly, we obtained 8% positivity of B19V, presenting concomitant infection with CHIKV in 17.1% of cases, 0.3% with ZIKV, 0.1% with DENV, 1% with DENV and CHIKV and 0.3% with CHIKV and ZIKV. Despite several seroprevalence studies [17–19], public attention to B19V in Brazil is limited. To our knowledge, B19V is rarely tested or even diagnosed in Public Health Centers. Importantly, from a clinical perspective, the disease presents with fever, polyarthropathy and rash and is often confused with pathogens such as DENV, CHIKV or ZIKV. B19V is the causative agent of erythema infectiosum, a viral exanthem and common childhood rash giving the appearance of "slapped cheeks" [20]. Arbovirus infections occur throughout the year in Brazil, but explosive outbreaks tend to occur during the summer period between November and April [21–23], in a similar fashion to other outbreak reports of erythema infectiosum [24–28]. With simultaneous arboviral circulation, clinical diagnoses are often based on epidemiological grounds, with infective agents responsible for similar disease presentations overlooked. Our findings are consistent with this notion and corroborate with

an earlier report from Di Paola et al [29], where an outbreak of B19V was reported during a DENV epidemic. In our study, the presenting patient symptoms of fever, headache, myalgia, arthralgia, and exanthema were, respectively, the most frequently identified. Di Paola et al., 2019 report demonstrates a similar symptom profile.

Diagnosis and the subsequent clinical management and reporting of arboviral infections in co-endemic locations can be extremely complex and is generally based on clinico—epidemiological grounds; where a case is reported to the health authorities as suspected or confirmed based on the Ministry of Health case definitions, using clinical symptoms and blood test results, such as leukocyte and platelet counts [30]. In 2020, among DENV cases reported up to April, only 23% were tested in reference laboratories [2], corroborating the analyses presented in our study. Laboratory confirmations, although more robust, are consistent with the clinico-epidemiological criteria demonstrated, especially among suspected cases of CHIKV (6.3%), followed by DENV (2.2%) and ZIKV (0.7%). Cases determined as unknown, due to lack of information or incorrect filling of the SINAN form, presented a higher percentage among suspected cases of ZIKV (93.8%), while suspected cases of DENV and CHIKV presented 82.6% and 60.1%, respectively. In this context, similar diseases may not be properly identified, as demonstrated in our study and in the findings of Di Paola et al. [29], where an outbreak of B19V was reported amidst a DENV epidemic. Fever, headache, myalgia, arthralgia and exanthema were, respectively, the most frequently found symptoms among individuals with samples analysed in our study, demonstrating some crossover with the report by Di Paola et al [29].

Studies show that the overall prevalence of B19V infections is derived from reported seropositivity rates; where in developed countries, seropositivity increases with age. Approximately 5–10% for children aged 2 to 5 years, 50% by age 15, and 60% will have evidence of prior B19V infection by age 30 [31–33]. In contrast, our analyses showed predominance for B19V infections in the age groups between 20–39 years; with 89% female and 61.8% of those pregnant, strongly indicating a first infection by the virus. This is surprising, since exposure to B19V typically occurs during childhood [34]. For non-immune pregnant women, B19V infections during seasonal epidemics can be as high as 10% [30,33,35] Although B19V is generally considered a minor, self-limiting infection in children and adults, a fetal infection can have serious congenital consequences. Despite a low seroconversion rate of 1% to 2% in non-immune women of childbearing age, the incidence of vertical transmission through placenta to the foetus can vary from 17% to 33% [31,36,37]. Congenital complications include hydrops fetalis and fetal anaemia–both of which can be diagnosed with non-invasive ultrasound. Intrauterine transfusion for treatment of severe foetal anaemia is typically successful and well tolerated. However, the sinister fetal complications of miscarriage and stillbirth appear to be highest when the primary infection occurs in the first half of pregnancy. In women who are less than 20 weeks into their pregnancy, miscarriage rates can range from 8% to 17%, with approximately 13% occurring in the first trimester [33,37,38]. Fortunately, adverse outcomes of acute fetal infection are rare and diminish as the pregnancy progresses [38]. Beyond clinical considerations in the pregnant host, there remains a well-documented association between adult B19V infection and myocarditis. DENV, CHIKV and ZIKV are each potential causes of cardiovascular illness, including myocarditis. Therefore, epidemiological diagnosis of arboviral infections in the backdrop of an epidemic in patients who may go on to develop myocarditis, again, may not always be correct and diagnostic certainty would benefit from molecular testing. Such data should feed back into the formation of appropriate public health strategies and to quantify the success of intervention throughout an epidemic response.

Ultimately, our analyses have demonstrated that passive surveillance measures may be vulnerable to failed detection of milder pathogens such as B19V, which in turn also raises concerns that we may be overlooking alternative pathogens, with unclear clinical consequences.

## Conclusion

The molecular tests conducted in this study proved effective in confirming the presence of the primary arboviruses circulating in Brazil and detecting concurrent infections by other viruses. This suggests their usefulness in diagnosing individuals with undefined etiologies, particularly during seasonal outbreaks of arboviral illnesses accompanied by acute febrile symptoms. Our findings should contribute to hospital and laboratory surveillance efforts, with an emphasis on co-circulating pathogens, such as B19V, which poses a significant risk of congenital abnormalities or miscarriage during the early stages of pregnancy. On a practical note, throughout epidemics of suspected arboviral infections, testing the pregnant patient presenting with fever, arthralgia and rash for B19V infection as part of the differential work-up would prompt diagnostic investigation with non-invasive ultrasound for hydrops fetalis and severe fetal anaemia; and thereby have the potential to expedite fetus saving treatment.

The main limitations of this study are the use of secondary databases, which is subject to errors in filing and classification, underreporting of cases and changes in the variables collected by SINAN. The conclusions of ecological studies refer to aggregates of data and may not be valid at the individual level.

## Acknowledgments

We thank the Centre for Arbovirus Discovery, Diagnostics, Genomics & Epidemiology (CADDE), Central Public Health Laboratory (LACEN-RN) and the Guamare Health Department for their support in the production of this manuscript. To the Ministry of Health for donating the reagents used in this study.

## Author Contributions

**Conceptualization:** Vanessa dos Santos Morais.

**Data curation:** Vanessa dos Santos Morais, João Felipe Bezerra, Flavia Emmanuelle Cruz, Themis Rocha de Souza, Antonio Charlys da Costa.

**Formal analysis:** Vanessa dos Santos Morais, Lídia Maria Reis Santana.

**Funding acquisition:** Ester Cerdeira Sabino, Antonio Charlys da Costa.

**Investigation:** Vanessa dos Santos Morais, Antonio Charlys da Costa.

**Methodology:** Vanessa dos Santos Morais, Rafael Augusto Alves Raposo, Roberta Marcatti, Erick Matheus Garcia Barbosa, Antonio Charlys da Costa.

**Project administration:** Vanessa dos Santos Morais, Lídia Maria Reis Santana, Antonio Charlys da Costa.

**Resources:** Vanessa dos Santos Morais, João Felipe Bezerra, Flavia Emmanuelle Cruz, Themis Rocha de Souza, Ester Cerdeira Sabino, Antonio Charlys da Costa.

**Software:** Vanessa dos Santos Morais, Lídia Maria Reis Santana.

**Supervision:** Ester Cerdeira Sabino, Antonio Charlys da Costa.

**Validation:** Vanessa dos Santos Morais, Antonio Charlys da Costa.

**Visualization:** Vanessa dos Santos Morais, Antonio Charlys da Costa.

**Writing – original draft:** Vanessa dos Santos Morais, Lídia Maria Reis Santana.

**Writing – review & editing:** Vanessa dos Santos Morais, Roozbeh Tahmasebi, Philip Michael Hefford, Renata Buccheri, Ester Cerdeira Sabino, Antonio Charlys da Costa.

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
