## [Decision Letter · Decision Letter 0]

9 Feb 2023

Dear Dr MORAIS,

Thank you very much for submitting your manuscript "Diagnostics and Molecular Characterization during an outbreak of Chikungunya virus and Primate Erythroparvovirus 1 in individuals with acute febrile illness in Rio Grande do Norte, Northeast of Brazil in 2016" for consideration at PLOS Neglected Tropical Diseases. As with all papers reviewed by the journal, your manuscript was reviewed by members of the editorial board and by several independent reviewers. In light of the reviews (below this email), we would like to invite the resubmission of a significantly-revised version that takes into account the reviewers' comments. 

We cannot make any decision about publication until we have seen the revised manuscript and your response to the reviewers' comments. Your revised manuscript is also likely to be sent to reviewers for further evaluation.

Sincerely,

Gregory Gromowski

Academic Editor

Abdallah Samy

Section Editor

Reviewer's Responses to Questions

**Key Review Criteria Required for Acceptance?**

**Methods**

-Are the objectives of the study clearly articulated with a clear testable hypothesis stated?

-Is the study design appropriate to address the stated objectives?

-Is the population clearly described and appropriate for the hypothesis being tested?

-Is the sample size sufficient to ensure adequate power to address the hypothesis being tested?

-Were correct statistical analysis used to support conclusions?

-Are there concerns about ethical or regulatory requirements being met?

Reviewer #1: In this draft, entitled "Diagnostics and Molecular Characterization during an outbreak of Chikungunya virus

and Primate Erythroparvovirus 1 in individuals with acute febrile illness in Rio Grande

do Norte, Northeast of Brazil in 2016", authors describe epidemiological data regarding the urban arbovirus Dengue, Zika and Chikungunya; however, only results of molecular diagnosis are described, and further characterization are missing. This is more a risk associated study, and authors should reconsider changing the title, or include more data about the viruses studied.

Reviewer #2: In this article, Vanessa dos Santos Morais et al. described a retrospective analysis of sera from acute febrile illness during year 2016 the time of CHIKV and ZIKV epidemic. Differential diagnosis were performed by PCR for CHIKV, ZIKV, Dengue and surprizingly for other viruses with generic system to flavivirus, alphaviruses enterovirus and parvovirus B19.

The abstract correspond to a single descriptive analysis and a short background about the reason that lead to analyse the prevalence of B19V was not presented except in the conclusion. This might be of particular interest considering the fact that Brazil was the country in which the level of fetal microcephaly was 10 or 20 times upper than the one found in the other countries. following the ZIKV epidemic that followed the CHIKV epidemic. Thus close to 6 years later, the question remained open about the cause to such a so high defect. 

These point should be clarified taking in account the result obtained. 

To note BV19 is a DNA virus thus the diagnosis procedure is not clear here. In general the procedure description deserved to be completed and the single reference to previous published article is not sufficient.

IN addition, the first part of the result swas the description of the arbovirus epidemic that take in account the 86 628 cases of putative infections. This must be indicated in the abstract and explain the choice of the molecularly analyzed 713 sera. 

In the method the available volume have to be indicated if we take in account the numbers of performed PCR and extraction. At the minima, there were two extractions one for the RNA viruses the other for the DNA viruses.

In addition despite a very large statistical analysis the author miss to explain the relationship between the molecular analysed samples and the general positive samples.

The presentation of prematurity data in pregnant women is not explained !!

Minor comments:

In table 1 and 2 there was a mix between English and Portuguese language.

Reviewer #3: See general comment

Detailed comments:

Page 7, line 209: delete bracket with virus list as this is already

listed in the cited reference Patel et al

Rephrase to:...for the detection of Flaviviruses, Alphaviruses and

Enteroviruses as described in Patel et al., Giry et al., and Hymas et

al., respectively (10-12)

Page9, line 274: ...on the condition...

**Results**

-Does the analysis presented match the analysis plan?

-Are the results clearly and completely presented?

-Are the figures (Tables, Images) of sufficient quality for clarity?

Reviewer #1: Most tables are difficult to understand, as they are in portuguese. Categories in columns must be redone. Please, also put maps with the same size and the same scale size. Check figure 6. Also, in figure 6, inform which map corresponds to DENV, CHIKV and ZIKV. Also, was B19 associated with any malformation? these information are poorly described.

Reviewer #2: No

Reviewer #3: See general comment

Detailed comments:

Table 1 & 2 , column 1: please translate to english, add legend

explaining the %value in the brackets in columns 2 & 3

Page 11, line 327: Here you are talking about the disease not the virus

and if the value sin in table 1 are correct izts actually:

"Higher among Chikunguya suspects and lower among Dengue and Zika fever

suspects"

Page 11, line 336: Please rephrase - unclear to me

Page 13, line 349:

Regarding arbovirus infections of pregnant women, 2,640 cases were

considered probable in the SINAN database, 1.8% for DENV, 4.0% for CHIKV

and 15.9% for ZIKV respectively and the highest percentage was observed

in the age groups between 20-29 years, being 50.3% for CHIKV, 50.1% 354

for DENV and 51.2% for ZIKV (Table 2).

Page 16: the graphs are double in 2 difrent sizes, remove one set

please.

Page 17, line 387: of which samples ? ...all samples or samples from

pregnant women ? Which group do the n= 492 represent ?

Page 17, line 389: List First double arbovirus infections , then

arbovirus +B19, then Doble arbovirus +B19 infections Percentage is

missing for CHIKV+B19

Page21, line 464 ...and ZIKV infection....

Something is missing in the sentence please rephrase.

**Conclusions**

-Are the conclusions supported by the data presented?

-Are the limitations of analysis clearly described?

-Do the authors discuss how these data can be helpful to advance our understanding of the topic under study?

-Is public health relevance addressed?

Reviewer #1: The authors conclude that more tests should be performed, which is largely known for many Brazilian states. Also, limitations regarding risk analysis based on probable cases should also be considered, as only a few samples were sent for diagnosis by PCR.

Reviewer #2: There was a large lack of structure in this article.

Reviewer #3: See general comment

Detailed comments

page 25, line 559: If These figures are the same as on page 17 line 387

then please heck the percentage for CHIKV & DENV in both paragraphs.

page 25, line 566: Why did you test for B19 ? Because of the issue xpu

know it can cause in pregnancy or becaus eof DiPaola`s study. Ther is no

problem in clearly saying well we saw DiPaola´s study and we wanted to

include this anaylses to see if we could falsify / confirm what was

deescribed

**Editorial and Data Presentation Modifications?**

Reviewer #1: (No Response)

Reviewer #2: the article should be largely rewritten to provide an article interesting to be read and to highlight the obtained results

Reviewer #3: (No Response)

**Summary and General Comments**

Reviewer #1: Morais and colleagues provide some epidemiological analysis regarding the main urban arboviruses in Rio Grande do Norte, Brazil. However, there are no viruses characterization, as the manuscript is entitled, and therefore new analyses must be done. I also have some minor points I hope it will improve the manuscript.

line 140: qPCR and RT-qPCR, as these arboviruses are RNA viruses.

line 209: there's no need to list all the virus species the PCRs can detect.

Reviewer #2: The article should be largely rewritten to provide an article interesting to be read and to highlight the obtained results.

In the current form there is a large lack of interest and scientific proof of the obtained results as the molecular methods were poorly described.

Reviewer #3: Altogether this is an interesting study worth publishing. However there

are some structural problems with the paper and some language

issues.

The motivation to look for premature births is not well explained in the

Introduction and Results. The first mention of the effect ZIKV on

pregnancy being the driver for the study analyses is in the discussion

in line 555.

The Introduction should set out the motivation and hypotheses that

detemined the design of the study based on existing literature. 

The motivation should be refered to in the

appropriate sections of the results.

The discussion should be sharpened and explain how the results confirm or dispel the hypothesis.

Qzuick search of literature on CHIKV / arboviruses and premature birth

https://pubmed.ncbi.nlm.nih.gov/34448947/

https://pubmed.ncbi.nlm.nih.gov/30169203/

PLOS authors have the option to publish the peer review history of their article (what does this mean?). If published, this will include your full peer review and any attached files.

Reviewer #1: No

Reviewer #2: No

Reviewer #3: No
---

## [Decision Letter · Decision Letter 1]

27 Jul 2023

Dear Dr MORAIS,

Thank you very much for submitting your manuscript "Detection of coinfection with Primate Erythroparvovirus 1 and arboviruses (DENV, CHIKV and ZIKV) in individuals with acute febrile illness in the state of Rio Grande do Norte in 2016" for consideration at PLOS Neglected Tropical Diseases. As with all papers reviewed by the journal, your manuscript was reviewed by members of the editorial board and by several independent reviewers. The reviewers appreciated the attention to an important topic. Based on the reviews, we are likely to accept this manuscript for publication, providing that you modify the manuscript according to the review recommendations. 

Sincerely,

Gregory Gromowski

Academic Editor

Abdallah Samy

Section Editor

Reviewer's Responses to Questions

**Key Review Criteria Required for Acceptance?**

**Methods**

-Are the objectives of the study clearly articulated with a clear testable hypothesis stated?

-Is the study design appropriate to address the stated objectives?

-Is the population clearly described and appropriate for the hypothesis being tested?

-Is the sample size sufficient to ensure adequate power to address the hypothesis being tested?

-Were correct statistical analysis used to support conclusions?

-Are there concerns about ethical or regulatory requirements being met?

Reviewer #2: The objective of the study is more or less clearly presented and should be highlighted in the abstract and at the end of the introduction or background. To note the abstract is quite too long.

The study design however is appropriate but the presentation of the samples tested in this study is not clear. In fact what we understand is that the 713 samples were NOT tested by molecular method before the present study and not that they were negative by a molecular test before to be send to lab. 

So the population is finaly not well described in the abstract (even if many details were given in the results)

To note this is not clear who performed the molecular test and a major weakness of the paper is the lack of details of this molecular test both the one that give positive results and those that were negative. Details (materials, primers, positive and negative controls must be given).

Finaly there was no clear hypothesis thus I cannot conclude about the sample size to test it.

The choice of the viruses tested (Flavivirus, alphavirus, enterovirus or BV19) MUST be explained in the introduction and not only in the discussion.

I imagine that the hypothesis that support this paper is “as the patient are negative for the test, it might be another pathogens (virus ?) that induced the disease”. 

Here the statistical test question is not appropriate.

The ethical and regulatory requirement are meet in this article.

Reviewer #3: -Are the objectives of the study clearly articulated with a clear testable hypothesis stated?

yes

-Is the study design appropriate to address the stated objectives?

yes

-Is the population clearly described and appropriate for the hypothesis being tested?

yes

-Is the sample size sufficient to ensure adequate power to address the hypothesis being tested?

yes

-Were correct statistical analysis used to support conclusions?

yes

-Are there concerns about ethical or regulatory requirements being met?

no

**Results**

-Does the analysis presented match the analysis plan?

-Are the results clearly and completely presented?

-Are the figures (Tables, Images) of sufficient quality for clarity?

Reviewer #2: Yes the analysis presented match the objective (if I have understand it). The result are clearly presented as well as the figures.

Reviewer #3: -Does the analysis presented match the analysis plan?

yes

-Are the results clearly and completely presented?

yes

-Are the figures (Tables, Images) of sufficient quality for clarity?

yes

**Conclusions**

-Are the conclusions supported by the data presented?

-Are the limitations of analysis clearly described?

-Do the authors discuss how these data can be helpful to advance our understanding of the topic under study?

-Is public health relevance addressed?

Reviewer #2: The conclusion is supported by the data but the reason to the choice of the virus tested is not clearly assessed and must be given in the introduction after the background and moreover the methods used fully described with postivite and negative control, cut-off of postive response, volume of the samples used and also is there are whole blood, blood spot on paper, sera etc.. 

The limitaions are clearly described as well as the usage of them for knowledge advance. BUT one more time the methods to obtain the result are not discussed.

Finaly, the public health relevance is addressed.

Reviewer #3: -Are the conclusions supported by the data presented?

yes

-Are the limitations of analysis clearly described?

yes

-Do the authors discuss how these data can be helpful to advance our understanding of the topic under study?

yes

-Is public health relevance addressed?

yes

**Editorial and Data Presentation Modifications?**

Reviewer #2: The abstract is too long and the thru objective need to be clearly stated

Reviewer #3: The paper is much improved

The following minor edits a re still required

Italics only if you use the taxonomic designation not if you generalize !

page 8, line 247 ....of Flaviviruses, Alphaviruses and Enteroviruses. 

page 13, line 360 ... for other Flaviviruses, Alphaviruses and Enteroviruses showed negative results...

page 16, line 445 ...After identifying DENV, CHIKV and ZIKV, we proceeded to test for other Flaviviruses, Alphaviruses and Enteroviruses, as well as for B19V...

page 16, line 447 ...In our analysis, there were no additional identifications of Flaviviruses and Alphaviruses other than DENV, ZIKV and CHIKV.

page 12, line 330 ... you chose not to use the suggested phrasing of the first review which was: 

Regarding arbovirus infections of pregnant women, 2,640 cases were

considered probable in the SINAN database, 1.8% for DENV, 4.0% for CHIKV

and 15.9% for ZIKV respectively and the highest percentage was observed

in the age groups between 20-29 years, being 50.3% for CHIKV, 50.1% 354

for DENV and 51.2% for ZIKV (Table 2).

You new paragraph however is again not well written and I suggest you use the suggested paragraph

above after all, or ask any other native speaker to help you repharse the paragraph which cannot stay

as it is.

page 13,line 355 ... showed positivity

**Summary and General Comments**

Reviewer #2: Detection of coinfection with Primate Erythroparvovirus 1 and arboviruses (DENV,CHIKV and ZIKV) in individuals with acute febrile illness in the state of Rio Grande do Norte in 2016.

In this new version of their article, Vanessa dos Santos Morais et al. answered to most of the questions and take in account the main suggestions. However, the description of the molecular analysis remained very poor. In the minima, the fact that the methods used were real time PCR should be given and not nested classicals. Even if the methods are already published some details about the targeted gene region, the usage of multiplex system etc… deserved also to be given. We know very well that most of the methods needs to be adapted or tested within a lab. Thus a short schapter explaining the quality control and the positive and negative control used as well as the cut-off chosen to consider a Ct as significant to give a positive result (i.e. detection of the DNA or RNA target) must be provided.

Thus mainly because all the paper conclusion is based on the molecular test value and quality….I cannot support the publication in the current form 

Minor comments: 

The abstract is too long, this is not an abstract but a copy past of some section of the paper.

Line 172-173 : this is not common to described in increase of 2016 compared to 2017 ! This must be a decrease of 2017 compared to 2016. The time direction is from the past to the future (We are not in “return to the future” movies).

Reviewer #3: (No Response)

PLOS authors have the option to publish the peer review history of their article (what does this mean?). If published, this will include your full peer review and any attached files.

Reviewer #2: No

Reviewer #3: No

Figure Files:

Data Requirements:

Reproducibility:

References

---

## [Editor Report · Decision Letter 2]

5 Oct 2023

Dear Dr MORAIS,

We are pleased to inform you that your manuscript 'Detection of coinfection with Primate Erythroparvovirus 1 and arboviruses (DENV, CHIKV and ZIKV) in individuals with acute febrile illness in the state of Rio Grande do Norte in 2016' has been provisionally accepted for publication in PLOS Neglected Tropical Diseases.

Best regards,

Gregory Gromowski

Academic Editor

Abdallah Samy

Section Editor

---

## [Editor Report · Acceptance letter]

27 Oct 2023

Dear Dr MORAIS,

We are delighted to inform you that your manuscript, "Detection of coinfection with *Primate Erythroparvovirus 1* and arboviruses (DENV, CHIKV and ZIKV) in individuals with acute febrile illness in the state of Rio Grande do Norte in 2016," has been formally accepted for publication in PLOS Neglected Tropical Diseases.

Best regards,

Shaden Kamhawi

co-Editor-in-Chief

Paul Brindley

co-Editor-in-Chief
